# Spatiotemporal Correlation Analysis of Hydraulic Fracturing and Stroke in the United States

**DOI:** 10.3390/ijerph191710817

**Published:** 2022-08-30

**Authors:** Chuanbo Hu, Bin Liu, Shuo Wang, Zhenduo Zhu, Amelia Adcock, James Simpkins, Xin Li

**Affiliations:** 1Lane Department of Computer Science and Electrical Engineering, West Virginia University, Morgantown, WV 26505, USA; 2Department of Management Information Systems, West Virginia University, Morgantown, WV 26505, USA; 3Department of Radiology, Washington University, St. Louis, MO 63110, USA; 4Department of Civil, Structural and Environmental Engineering, University at Buffalo, Buffalo, NY 14260, USA; 5Department of Neurology, West Virginia University, Morgantown, WV 26505, USA; 6Department of Neuroscience, West Virginia University, Morgantown, WV 26505, USA

**Keywords:** fracking, stroke mortality, annualized loss expectancy (ALE), geographically and temporally weighted regression (GTWR), spatiotemporal analysis

## Abstract

Hydraulic fracturing or fracking has led to a rapid growth of oil and gas production in the United States, but the impact of fracking on public health is an important but underresearched topic. We designed a methodology to study spatiotemporal correlations between the risk of fracking and stroke mortality. An annualized loss expectancy (ALE) model is applied to quantify the risk of fracking. The geographically and temporally weighted regression (GTWR) model is used to analyze spatiotemporal correlations of stroke mortality, fracking ALE, and nine other socioeconomic- and health-related factors. The analysis shows that fracking ALE is moderately correlated with stroke mortality at ages over 65 in most states of fracking, in addition to cardiovascular disease and drug overdose being positively correlated with stroke mortality. Furthermore, the correlations between fracking ALE and stroke mortality in men appear to be higher than in women near the Marcellus Shale, including Ohio, Pennsylvania, West Virginia, and Virginia, while stroke mortality among women is concentrated in the Great Plains, including Montana, Wyoming, New Mexico, and Oklahoma. Lastly, within two kilometers of the fracking mining activity, the level of benzene in the air was found to be significantly correlated with the fracking activity in Colorado.

## 1. Introduction

Hydraulic fracturing [1], also known as fracking, is a geochemical engineering process by which large volumes of water combined with chemical and sand proppants are injected into tight formations with high pressure to fracture and facilitate recovery of unconventional reserves of oil and gas [2]. With the development of fracking technology in the United States, shale gas is becoming an increasingly important source of natural gas, and interest has spread to countries around the world with shale gas storage. The US Energy Information Administration (EIA) assessed 137 shale formations in 42 countries around the world [3], and the distribution of assessed shale gas and shale oil basins of the world is shown in Figure 1. They represent 10% of the world’s crude oil and 32% of the world’s natural gas. However, as of 2013, only three countries (United States, Canada, and China) have significant commercial shale gas production due to technical limitations and local laws.

Currently, the United States is the world’s largest producer of both natural gas and crude oil. According to the US Energy Information Administration (EIA), there were approximately 23,000 fracking wells in the US in 2000. In 2015, the number of fracking wells increased rapidly to approximately 300,000, representing 67% of United States natural gas production and 51% of United States crude oil production [4]. Despite the economic benefits of fracking, this expansion has brought the industrial activities of oil and gas development closer to backyards and communities, increasing the risk of human exposure to new contaminants and threats [5,6].

The health implications and effects of fracking have not been adequately studied [7,8,9,10,11,12]. There is a growing body of research studies on the negative impact of fracking on air and water quality [13,14,15,16,17], as well as public health [18,19,20,21,22]. Although health discussions have focused on drinking water contamination, particularly in the eastern US, there is growing interest in studying a variety of health threats arising from air pollution [7,23]. Health threats from air pollution vary significantly across environments [24,25,26,27]. For example, research has attempted to link fracking pollution with unhealthy levels of smog and toxic air pollutants [28]. Exposure to this pollution can cause eye, nose, throat, respiratory disease, birth defects, cancer, or premature death [29]. However, little is known about whether fracking can cause life-threatening conditions such as stroke [8].

Stroke as a neurological disease is the leading cause of long-term adult disability and the fifth leading cause of death in the United States [30], with approximately 795,000 stroke events annually. Stroke belt refers to a consistent pattern of striking geographic variation in stroke mortality rates within the United States [31]. It covers 11 states in the southeast US with an unusually high incidence of stroke and other forms of cardiovascular disease. Factors that explain the prevalence of excessive stroke in the stroke belt include differences in socioeconomic status (e.g., employment rate and marital rate), risk factors (e.g., smoking and unhealthy diet), and prevalence of common chronic diseases (e.g., diabetes and heart disease) [32,33]. A recent study has shown that the highest contributors to the Stroke Belt include a higher burden of risk factors, higher levels of inflammation and infection, and lower socioeconomic status, while environmental exposures and lifestyle choices are considered lesser contributors [34].

*Does fracking induce a higher risk of stroke?* Although a potential connection between fracking and stroke has been mentioned in the literature (e.g., [35,36,37]), there is no systematic study to address this question. The closest research to this work is the study on the impact of fracking on water pollution [38] and air pollution [39], but its research data are limited to the local area. Furthermore, the spatial extent of the public health impact of hydraulic fracturing is a question that existing research attempts to answer. For example, the distance to the nearest fracking well has been used as an important indicator to analyze the spatial correlation between fracking and infant health [40]. A spatial analysis method has been designed to quantify the environment at risk of Marcellus Shale fracking in the state of PA, USA [41]. The study [42] verified that people within 0.8 km of a fracking well are particularly at risk to their health. However, few studies have analyzed the spatiotemporal impacts of hydraulic fracturing on public health. The goal of this study is to address this question from a geographic information system (GIS) perspective using the extension of the geographically weighted regression (GWR) method [43]: geographically and temporally weighted regression (GTWR) [44]. We performed a detailed regression analysis of the stroke and fracking data using GTWR. We hope that this work can shed light on the relations between fracking and stroke risk and stimulate more quantitative studies on the health risk of fracking, which can better inform decision makers about energy and public health policy.

## 2. Methodology

### 2.1. Study Area and Data Collection

To study the spatiotemporal correlations of stroke mortality and fracking, the present study has chosen 49 states in the US as the study area. Alaska is not included due to its geographical isolation. Figure 2 shows the stroke death rate per 100,000 people over 65 years of age and all sites of fracking activity before 2018. To explore the impact of fracking on stroke mortality, we divide the 49 states into *fracking states* and *non-fracking states*. As a result, there are 24 fracking states that had fracking activities (including 19 states with active fracking and 5 states with little fracking) and 25 non-fracking states that did not have fracking activity by 2018.

Stroke is closely related to people’s behavior habits (tobacco use, high cholesterol diet, and physical activity index), socioeconomic status (family mean income, marital rate, and employment rate), and other diseases (cardiovascular, overdose, and diabetes) [33,45]. Therefore, these variables have been selected for comparison with risk factors for fracking. Table 1 shows the details of the dependent and explanatory variables and their data sources. All data was collected in the US from 2010 to 2018. The scale of the data we collected is state-level except for fracking, because the county-level data contain a lot of missing data that can cause problems with the analysis. Furthermore, since stroke is a chronic disease, we processed the fracking data into the cumulative number of fracking wells. In other words, the number of fracking wells in any year includes all fracking wells before that year.

### 2.2. Quantitative Risk Analysis: Annual Loss Expectancy of Fracking

Recent research illustrated that 2–16% of oil and gas wells spill liquids every year [46]. These accidents have caused fracking chemicals to contaminate drinking water and air, further raising serious public health risks [47,48,49]. To quantify the public health risk of fracking, the annualized loss expectancy (ALE) model [50] has been applied to assess the public health-related ALE caused by fracking by state, which is abbreviated as fracking ALE. ALE is the product of the annual rate of occurrence (ARO) and the single loss expectancy (SLE) [51] caused by fracking, as shown in Equation (Equation 1).
(1)ALE=ARO×SLE
where fracking ARO is the annual occurrence of pollutant leakage due to fracking, which is equal to the Fracking Density per square kilometer (FD) multiplied by the annual rate of occurrence per fracking per square kilometer (AROF), ARO = FD × AROF.

Fracking SLE is defined as a population with negative health impacts expected from the occurrence of fracking accidents. For risk calculation, SLE is used to calculate a single loss when a specific event occurs. Fracking SLE is calculated by multiplying the Population Density per square kilometer (PD) by the fracking exposure factor (EF), SLE = PD × EF, as the higher the population density in fracking-active states, the greater the negative impact of fracking on public health.

Fracking ALE represents the product of AROF, FD, PD, and EF, which means the average loss per year of environmental pollution caused by fracking on public health. The FD and PD of the *j*th state in the *i*th year are ${\mathrm{FD}}_{ij}$ and ${\mathrm{PD}}_{ij}$, respectively. As an important explanatory variable for the regression in Section 2.3.3, ALE is normalized as follows:(2)$$Normalized\;{\mathrm{ALE}}_{ij}=\frac{{\mathrm{ALE}}_{ij}-Min\left(\mathrm{ALE}\right)}{Max(\mathrm{ALE})-Min(\mathrm{ALE})}$$
where $Normalized\;{\mathrm{ALE}}_{ij}$ is the normalized ALE in state *j* in year *i*. Since AROF and EF are constants, Equation (Equation 2) can be further formulated as
(3)$$Normalized\;{\mathrm{ALE}}_{ij}=\frac{{\mathrm{FD}}_{ij}\times {\mathrm{PD}}_{ij}-Min\left(\mathrm{FD}\times \mathrm{PD}\right)}{Max(\mathrm{FD}\times \mathrm{PD})-Min(\mathrm{FD}\times \mathrm{PD})}$$

### 2.3. Regression Model

#### 2.3.1. Multicollinearity

We performed multicollinearity diagnostics before applying the regression model, since several explanatory variables could be highly correlated. Multicollinearity means that there is a high linear correlation between several specific explanatory variables, which could lead to bias in explaining the significance and associations of other variables. We adopted the variance inflation factor (VIF) [52], a metric of the severity of multicollinearity, to eliminate this problem. Explanatory variables with a VIF greater than 10 are considered to cause multicollinearity and should be excluded from the model [53].

#### 2.3.2. Spatial Autocorrelation

As a commonly used spatial autocorrelation test, Moran’s I test represents the spatial autocorrelation of a single explanatory variable and can be expressed as [54].
(4)$$I=\frac{n}{\sum_{i=1}^{n}\sum_{j=1}^{n}w_{ij}}\frac{\sum_{i=1}^{n}\sum_{j=1}^{n}w_{ij}\left(y_{i}-\bar{y}\right)\left(y_{j}-\bar{y}\right)}{\sum_{i=1}^{n}{\left(y_{i}-\bar{y}\right)}^{2}}$$

Moran’s I ranges from −1 to 1. A higher positive value indicates that closer observations have more similar attribute values, whereas farther observations have more distinct attribute values, indicating spatial aggregation. Negative values represent a spatially distributed distribution and a zero value represents a spatially random distribution. The null hypothesis of the Moran’s I test indicates that the explanatory variables are spatially independent. It indicates that the Moran’s I is close to zero. The Z-score is used as a significant indicator to measure Moran’s I to verify the null hypothesis, whose formula is as follows [54].
(5)$$Z\left(I\right)=\frac{I-E(I)}{\sqrt{Var(I)}}$$
where E(I) and Var(I) are the expectation and standard deviation of Moran’s I, respectively. The significance level in this study is established as the *p*-value < 0.05.

#### 2.3.3. Geographical and Temporal Weighted Regression (GTWR)

Spatiotemporal data analysis provides a series of important tools to solve problems such as correlation analysis of spatiotemporal data, spatiotemporal pattern analysis, and spatiotemporal prediction problems [44,55,56,57,58,59]. To analyze the spatio-temporal correlation between risk factors (see Table 1) and stroke mortality, the GTWR model was selected as the regression model. Compared to traditional Geographically Weighted Regression (GWR) [43], which only considers spatial features, the GTWR model [44] considers the non-stationary effect in space and time. Therefore, it was adopted to explore the spatiotemporal heterogeneity of the influence of fracking on different stroke mortality (i.e., age-based and gender-based) under the constraint of spatiotemporal differences. By establishing a three-dimensional elliptical coordinate system (including time, longitude, and latitude) in which the temporal dimension is the vertical dimension, in addition to the two horizontal spatial dimensions (longitude and latitude), the model can describe the spatio-temporal influence via the regression coefficients corresponding to explanatory variables. The GTWR model is described as
(6)$$Y_{i}=\beta_{0}\left(u_{i},v_{i},t_{i}\right)+\sum_{k}\beta_{k}\left(u_{i},v_{i},t_{i}\right)X_{ik}+\varepsilon_{i}$$
where *u* and *v* represent longitude and latitude, respectively. $\beta_{0}\left(u_{i},v_{i},t_{i}\right)$ represents the intercept item of state with centroid at ($u_{i}$, $v_{i}$) in year $t_{i}$; *k* is the number of explanatory variables; $\beta_{k}\left(u_{i},v_{i},t_{i}\right)$ is the regression coefficient of the *k*th explanatory variable in year *t*; $X_{ik}$ is the *k*th explanatory variable. These explanatory variables are defined in Table 1.

The correctional values of the Akaike Information Criterion (AICc) [60] is an important metric and often used to select explanatory variables and determine the final model with the lowest AICc. In this study, the ArcGIS GTWR plugin was used to analyze the spatio-temporal correlation between stroke mortality and risk factors such as fracking ALE.

## 3. Results

### 3.1. Model Comparison

To solve the multicollinearity problem between multiple variables, the variables with a VIF greater than 10 (i.e., hypertension and obesity) were removed. The results of the VIF values of significant explanatory variables are given in Table A1 for stroke mortality over 65 years and Table A2 stroke mortality 45–64 in Appendix A. Additionally, Moran’s I statistics were calculated to determine whether the explanatory variables in Table A1 and Table A2 are spatially associated. The results of the Moran’s I test are given in Table A3 in Appendix A. The 10 selected variables with a *p*-value below 0.05 were included in the regression model, indicating that all variables are spatially autocorrelated.

Furthermore, to increase the significance of the regression variable, two explanatory variables (i.e., heavy drink and education) have been removed using stepwise selection based on AICc. Furthermore, a comparison with three baseline models was implemented, including Ordinary Least Squares (OLS) [61], Temporally Weighted Regression (TWR) [44] and GWR [43]) to evaluate the performance of the GTWR model [44]. As shown in Table 2. GTWR outperforms OLS, TWR, and GWR in model fitting, demonstrating that it better explains dependent variable stroke mortality. Taking the fracking state model as an example, the values of $R^{2}$ increase from 0.757 in the OLS model, 0.768 in the TWR model, and 0.933 in the GWR model, to 0.970 in the GTWR model. The AICc reduces from −401.129 in the OLS, −399.780 in the TWR model and −534.502 in the GWR model, to −564.090 in the GTWR model (the lower, the better). The explanatory power increases significantly as spatial information and temporal information are considered in the model. In the rest of the paper, we will only analyze the results of the GTWR model.

Estimates of regression coefficients for 65+ stroke mortality in fracking states and non-fracking states are given in Table 3 and Table 4. The results of both models showed the positive effect of cardiovascular and overdose on 65+ stroke mortality, and the negative effect of marital rate and employment rate on 65+ stroke mortality. These results share some similarities with other findings of previous work [62,63,64]. Furthermore, Table 3 shows that most fracking ALE coefficients are positively associated with stroke mortality at 65 years, although the correlation was much lower than for stroke mortality variables such as cardiovascular disease and overdose. Table 3 and Table 4 show that there is no positive correlation between high cholesterol and stroke mortality. The result shares some similarities with other observations from previous work [33]. Furthermore, there is no correlation between tobacco use and 65+ stroke mortality, as its *p*-value > 0.1 (see Table A1).

### 3.2. Spatiotemporal Features of Fracking ALE Coefficients

We analyze the temporal and spatial characteristics of fracking ALE using the average values of the regression coefficients, which help to explore the temporal trends and spatial differences of fracking ALE on stroke mortality at ages 65 and older.

#### 3.2.1. Temporal Features of Fracking ALE Coefficients

The aforementioned improvement of GTWR is extended by incorporating the temporal dimension into the traditional GWR model. From the results of the GTWR model, we can obtain the time series of the yearly fracking ALE coefficients. Figure 3 presents the fluctuation of the average coefficients of the Fracking ALE variables over a 9-year period (from 2010 to 2018). Negative coefficients indicate the reverse correlation between the dependent and explanatory variables, and vice versa. 19 states with active fracking activity were discussed and 5 states with very few fracking activities were removed [65]. As Figure 3 shows, the positive correlation between fracking ALE and stroke mortality (65+) in California, Utah, Alabama, Louisiana, and Oklahoma decreased significantly year by year. In contrast, the positive correlation between fracking ALE and stroke mortality (65+) in North Dakota increased significantly. The coefficient of Fracking ALE in West Virginia, Ohio, Pennsylvania, Michigan, and Virginia slowly decreases. The ALE coefficient for fracking in Colorado and New Mexico first increases and then decreases.

#### 3.2.2. Spatial Features of Fracking ALE Coefficients

An important feature of the GTWR model is that the estimated coefficients are mappable for visual analysis. The spatial distributions of the effects of explanatory variables on ALE fracking are visualized in Figure 4. This study sets zero as a threshold to distinguish positive and negative effects. Darker states on the map have stronger positive correlations between fracking ALE and stroke mortality above 65. Figure 4 shows that the positive correlation between the ALE of fracking and stroke mortality (65+) in North Dakota, Ohio, Montana, Kansas, and Arkansas is stronger than in other states of fracking. Fracking has been active in North Dakota and Ohio [65]. Although fracking is generally active in Montana, Kansas, and Arkansas, fracking in their respective neighbors (e.g., North Dakota, Colorado, Texas) is always active. In contrast, the positive correlation in Virginia, Mississippi, and Oklahoma is weaker than in other states of fracking. Fracking is not active in the states of Virginia and Mississippi, according to the report by Environment America [4]. Additionally, Oklahoma, New Mexico, and Texas, where fracking is the most prevalent, have lower ALE coefficients for fracking than most states. The possible reason for this is that Fracking ALE is positively correlated with some socioeconomic factors, such as family mean income and marital rate (see Figure A1), which are negatively correlated with stroke mortality over 65 years (see Figure A2).

### 3.3. Comparative Analysis on the Effect of Fracking on Gender-Based Stroke Mortality

To explore spatio-temporal differences in the effect of hydraulic fracturing on gender-based stroke mortality, we performed spatio-temporal regressions on male and female stroke mortality separately using the selected explanatory variables. Based on the regression coefficients obtained for the two groups, the temporal and spatial characteristics of different sexes were compared and analyzed.

Figure 5 and Figure 6 show, respectively, the regression coefficients of fracking ALE for different dependent variables (male stroke mortality and female stroke mortality). We found that the fracking ALE regression coefficients for males with stroke mortality and females with stroke mortality have similar temporal trends in Ohio, West Virginia, Virginia, and Pennsylvania. They slowly decreased from 2010 to 2018. The correlation coefficient between fracking ALE and stroke mortality (both men and women) in California decreased significantly year by year. The correlation coefficient between fracking ALE and stroke mortality (both male and female) in North Dakota increased significantly year by year. The correlation coefficients in the state of Colorado first increased and then decreased for male stroke mortality, but reversed for female stroke mortality. Additionally, the correlation coefficients between ALE from fracking and stroke mortality (both male and female) in California and some states (Arkansas, Louisiana, Mississippi, and Alabama) in the stroke belt are lower than those of other states from fracking.

In addition to the temporal characteristics of gender-based stroke mortality, the fracking ALE regression coefficients are spatially differentiated (see Figure 7). The ALE coefficients of fracking for male and female on stroke mortality around Marcellus shale (including Pennsylvania, West Virginia, and Ohio) and New Mexico and Oklahoma are higher than those of other states of fracking. In contrast, the ALE coefficients for fracking on female stroke mortality are higher in Montana and Wyoming, but its ALE coefficients for male are lower.

## 4. Discussion

### 4.1. Does Fracking Cause a Higher Risk of Stroke?

Despite the potential health risks associated with fracking, there have been several quantitative studies on how fracking can affect public health on a local scale [40,66], characterizing the risk of fracking as the distance from the patient’s residence to the nearest well. However, these methods are based on privacy-protected clinical data that contain large amounts of patient personal information. They are not applicable to this study, since patient personal information is not included in the publicly available dataset from CDC. This study uses ALE to quantitatively study the possible connection between fracking and stroke using publicly available data. It can be observed from the results of the GTWR analysis that fracking has a non-negligible effect on stroke mortality above 65 in most areas with fracking prevalent, as shown in Table 3 and Figure 4. According to USCB, there were 40.3 M US residents 65 years and older in the 2010 Census and 54.1 M in the population estimates of 1 July 2019, (https://www.census.gov/topics/population/older-aging.html). With the aging of the United States and the increase in fracking activities, how to keep fracking activities away from communities with a high proportion of adults 65 years or older is a question that policy makers should consider. However, its impact is relatively minor compared to other more dominant factors, such as cardiovascular disease and overdose.

### 4.2. Spatiotemporal Differences in the Effect of Fracking on Age/Gender-Based Stroke Mortality

Deeper reasoning is needed to understand the geographic variations of cases of stroke mortality related to fracking based on the age and sex of the patients. As shown in Table A1 and Table A2, the Fracking ALE variable for stroke mortality between 45–64 years is not considered statistically significant with a *p*-value > 0.1, but the Fracking ALE variable for stroke mortality over 65 years is considered statistically significant with a *p*-value < 0.01. For gender-based stroke mortality, the higher Fracking ALE coefficient for men than for women seems to suggest that men are at higher risk, at least in regions near Marcellus shale. Additionally, higher fracking ALE coefficients for women than for men suggest that women are at greater risk in some states, including Montana, Wyoming, Oklahoma, and New Mexico. Furthermore, the fracking ALE coefficients for both male and female stroke mortality are small in some states of the stroke belt (e.g., Arkansas, Louisiana, Mississippi, and Alabama), suggesting that fracking is not a major factor in stroke mortality compared to some major factors, such as cardiovascular disease.

### 4.3. Air Pollutant Emissions from Fracking

We examined a possible mechanism for how fracking threatens public health. An existing report indicates that fracking produces environmental pollution, including hazardous water pollutants and hazardous air pollutants (HAPs) [67]. To explore which HAPs are associated with fracking activities, we collected air pollutants data from the HAP monitoring station in Colorado (longitude: −108.053259 and latitude: 39.453654). We have drawn a zone of interest with a radius of two kilometers around the HAP monitoring station and calculated the number of fracking wells. Four HAPs (Butadiene, Benzene, Formaldehyde, and Acetaldehyde) were monitored through this station, and we upsampled the four HAP monitoring data through linear interpolation to ensure that all four HAPs had the same time resolution. We then constructed a time series of fracking activities according to the start and end times of each fracking activity, and the overlapping fracking activities were aggregated. Finally, the four HAPs and the time series of the fracking activity were normalized, and then the fracking activities were analyzed for time series correlation with different HAPs using Pearson’s correlation [68]. Pearson’s r, which ranges from −1 to 1, was calculated to measure the degree of correlation between the two time series. When r > 0, a larger Pearson’s r suggests a stronger positive correlation. Table 5 shows that the Pearson correlation (*r*) of benzene was higher than the other three HAPs and the correlation is statistically significant (*p*-value <0.01). A study has shown the correlation between benzene exposure and the risk of cardiovascular disease due to the high level of trans, trans-muconic acid (t, t-MA) [69]. Although benzene exposure has also been found to be associated with high cholesterol [70], cardiovascular disease has a stronger correlation with stroke mortality at 65+ in our study than high cholesterol. The high density lipoprotein cholesterol was found to be more important for patients ≤65 years of age than older adults [71]. In addition, high cholesterol has been shown not to be associated with stroke mortality in some studies [33,72]. Therefore, the leakage of the chemical benzene due to fracking might contribute to cardiovascular disease thus stroke mortality but further study is needed and other processes such as water pollution might contribute as well.

### 4.4. What Is the Implication of This Study on Health Policy-Making?

With the increase in fracking activity, the socioeconomic environment, such as the employment rate and family income, continues to increase. At the same time, more and more people and communities in areas with many hydraulically fractured wells report health problems, such as cancer and harm to the nervous, respiratory, and immune systems. Macroscopically, we found that the effects of fracking and stroke mortality were not significant for the 19 active states of fracking in the US. This is likely due to the following reasons: (1) Fracking areas in the US are mainly distributed in the Great Plains and Marcellus Shale, which are often located in mountainous or suburban areas with low population density, which may lead to limited spatial impact. (2) The fracking process generally only lasts 3–5 days, which leads to a limited impact on time. (3) The development of fracking will promote local socioeconomic status (e.g., employment rate and marital rate), which is negatively related to stroke mortality. This may cause the effect of fracking on stroke mortality to be insignificant in some states (e.g., Texas) where fracking is active.

This study analyzed the correlation between fracking and different hazardous air pollutants based on the public air pollutant dataset from the US Environmental Protection Agency (EPA). We found that the concentration of benzene in the air was related to the calculated sequence of fracking activity in the buffer zone within a radius of 2 km (Pearson r=0.2452 and the *p*-value < 0.01). To minimize health risk, this research suggests that there should be no public facilities with a high population density within 2 km of fracking activities. Furthermore, the high concentration of benzene in the air may be due to fracking, which caused groundwater pollution due to the extremely high volatility of benzene. It may be important to identify and investigate domestic water wells that are within two kilometers of a fracking well. A more systematic study of the impact of fracking on water contamination [28] remains for future research.

## 5. Conclusions

This article provided a systematic study on the spatiotemporal correlation between fracking and stroke mortality using the GTWR model. The temporal trend of positive correlation between fracking ALE and stroke mortality shows a varying pattern from state to state. The spatial distribution appears to demonstrate that there is a gender difference between the Great Plains and the Marcellus Shale. Our regression results also show that disease-related risk factors, including cardiovascular and overdose, have a more significant correlation with stroke mortality over 65 years of age than those related to fracking. Finally, there appears to be a significant temporal dependency between fracking and air pollutant emissions, especially for benzene. Future studies may focus on developing county-level GWR/GTWR models, although missing data is a critical challenge to resolve.

## Figures and Tables

**Figure 1 ijerph-19-10817-f001:**
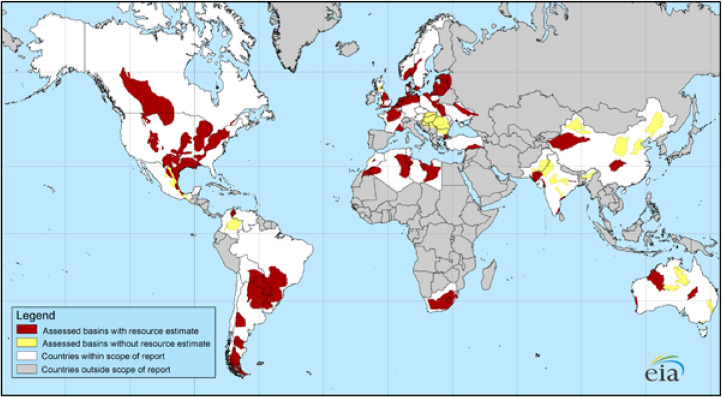
Mapping the distribution of assessed global shale gas and shale oil basins [3]. Source from United States Energy Information Administration (EIA) and United States Geological Survey.

**Figure 2 ijerph-19-10817-f002:**
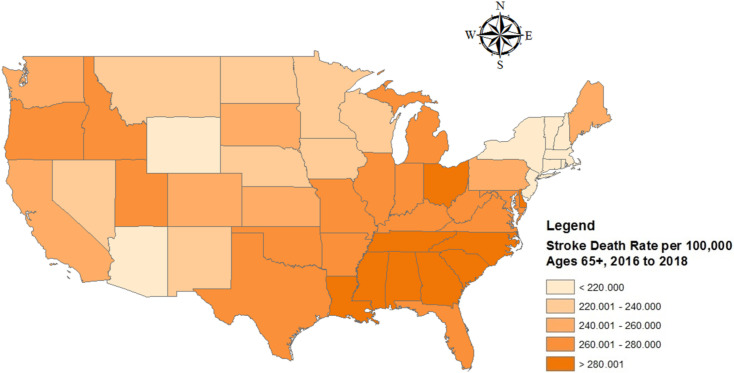
Map of the study area and the distribution of the stroke death rate per 100,000 population 65 years and older (**upper**) and US fracking activities by state prior to 2018 (**lower**).

**Figure 3 ijerph-19-10817-f003:**
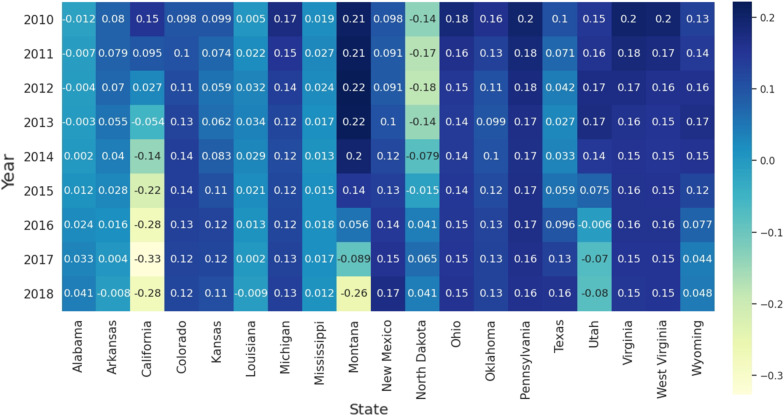
Temporal distribution of the average coefficients of the ALE of Fracking for 19 states with active fracking.

**Figure 4 ijerph-19-10817-f004:**
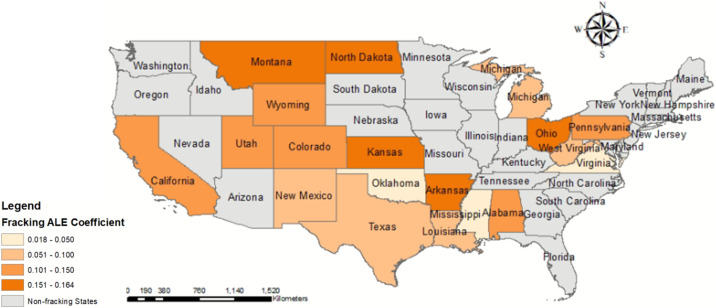
Spatial distribution of the average coefficients of fracking ALE for 19 states related to fracking.

**Figure 5 ijerph-19-10817-f005:**
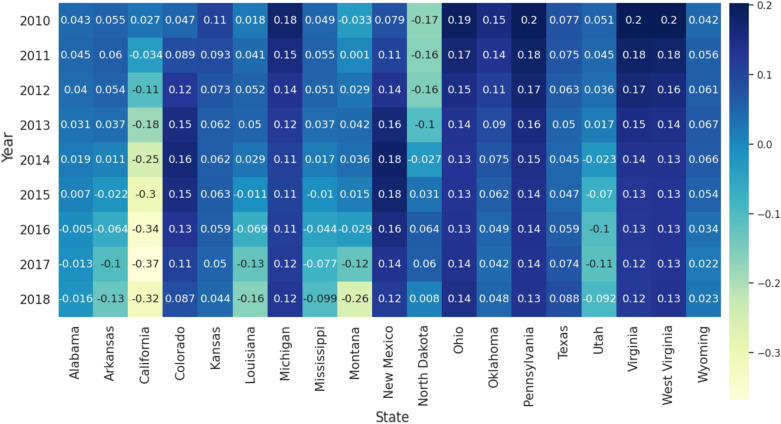
Temporal distribution of the average coefficients of the ALE of Fracking for 19 states related to fracking (Male).

**Figure 6 ijerph-19-10817-f006:**
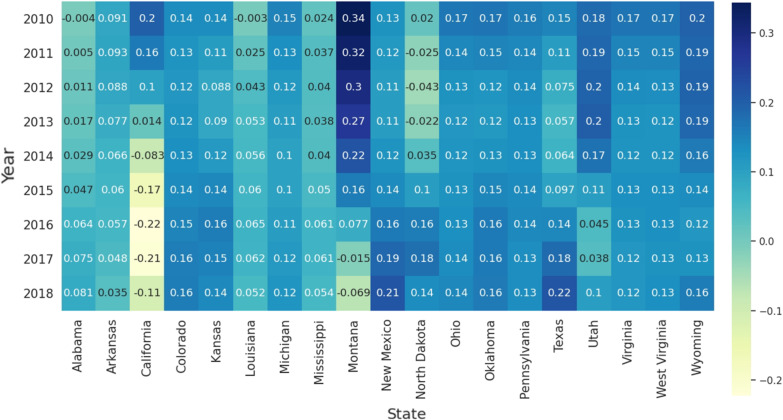
Temporal distribution of the average coefficients of ALE of fracking for 19 states related to fracking (Female).

**Figure 7 ijerph-19-10817-f007:**
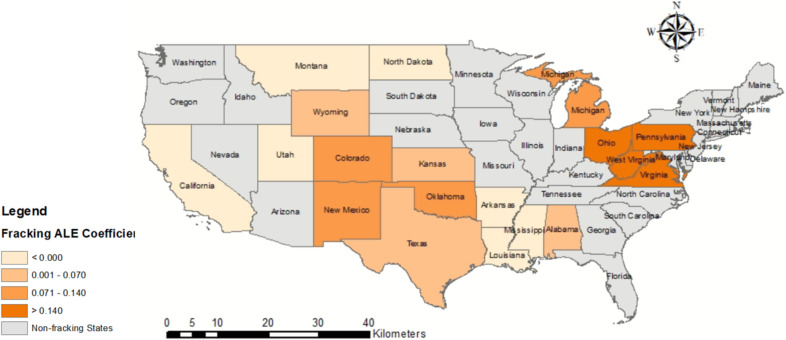
Spatial distribution of the average coefficients of fracking ALE for 19 states related to fracking for Male (**upper**) and Female (**lower**).

**Table 1 ijerph-19-10817-t001:** Description of the data used in this study (CDC: Centers for Disease Control and Prevention; USCB: The United States Census Bureau; FF: United States FracFocus; HAPs: Hazardous Air Pollutants; EPA: The United States Environmental Protection Agency).

Type	Variables	Description	Source
Fracking	Fracking activity	The location of Fracking wells	FF
Stroke mortality	65+ stroke mortality	Stroke deaths rate over 65 per 100,000	CDC
	Male stroke mortality	Male Stroke deaths rate per 100,000	CDC
	Female stroke mortality	Female Stroke deaths rate per 100,000	CDC
Disease	Diabetes	Proportion of diagnosed diabetes	CDC
	Cardiovascular	Cardiovascular deaths rate over 65 per 100,000	CDC
	Overdose	Drug overdose death rates	CDC
	Hypertension	High blood pressure deaths over 65 per 100,000	CDC
	Obesity	Adults with a BMI > 30	CDC
Behavior	Tobacco use	Current cigarette use by adults	CDC
	High cholesterol	High total cholesterol among adults	CDC
	Physical activity index	Physical Inactivity Prevalence	CDC
	Heavy Drink	8 or more drinks per week (female) or 15 or more drinks per week (male)	CDC
Socioeconomic	Mean income	Family income by number of workers	USCB
	Marital rate	Proportion of married population	USCB
	Employment rate	Proportion of employed population	USCB
	Education	Bachelor’s degree or higher	USCB
HAPs	Butadiene	Concentration monitoring data for Butadiene	EPA
	Benzene	Concentration monitoring data for Benzene	EPA
	Formaldehyde	Concentration monitoring data for Formaldehyde	EPA
	Acetaldehyde	Concentration monitoring data for Acetaldehyde	EPA

**Table 2 ijerph-19-10817-t002:** Comparison results of OLS, TWR, GWR, and GTWR models.

	Fracking States			Non-Fracking States		
	AICc	$R^{2}$	Adjusted $R^{2}$	AICc	$R^{2}$	Adjusted $R^{2}$
OLS [61]	−401.129	0.757	0.745	−293.313	0.601	0.584
TWR [44]	−399.780	0.768	0.757	−318.285	0.691	0.678
GWR [43]	−534.502	0.933	0.929	−481.016	0.897	0.892
GTWR [44]	**−564.090**	**0.970**	**0.968**	**−487.886**	**0.931**	**0.928**

**Table 3 ijerph-19-10817-t003:** Estimation of the GTWR model for 65+ stroke mortality in fracking states.

Variables	MIN	LQ	MED	UQ	MAX	AVG
**Related-disease risk factors**						
Diabetes	−0.661	−0.316	−0.001	0.218	0.632	−0.018
Cardiovascular	−1.506	−0.070	0.269	0.524	1.141	0.232
Overdose	−0.861	−0.030	0.389	0.693	1.148	0.347
**Behavior risk factors**						
Tobacco use	−0.973	−0.342	−0.157	−0.035	0.351	−0.197
High cholesterol	−1.346	−0.325	−0.149	0.090	0.415	−0.139
PAI	−0.585	−0.161	−0.080	0.020	0.521	−0.056
**Socioeconomic risk factors**						
Mean income	−1.442	−0.215	0.147	0.367	0.623	0.039
Marital rate	−1.459	−0.984	−0.677	−0.409	0.720	−0.663
Employment rate	−2.106	−0.604	−0.014	0.128	0.627	−0.223
**Fracking risk factor**						
Fracking ALE	−0.327	0.041	0.116	0.154	0.394	0.094
Intercept	−0.009	0.487	0.669	0.792	3.747	0.715

**Table 4 ijerph-19-10817-t004:** Estimation of the GTWR model for 65+ stroke mortality in non-fracking states.

Variables	MIN	LQ	MED	UQ	MAX	AVG
**Related-disease risk factors**						
Diabetes	−0.603	−0.085	0.227	0.314	0.467	0.109
Cardiovascular	−0.354	0.272	0.357	0.427	0.896	0.305
Overdose	−1.341	−0.189	0.166	0.286	0.805	0.037
**Behavior risk factors**						
Tobacco use	−0.577	−0.036	0.102	0.222	0.71	0.08
High cholesterol	−0.841	−0.539	−0.296	−0.165	0.175	−0.333
PAI	−0.493	−0.101	0.043	0.233	0.451	0.063
**Socioeconomic risk factors**						
Mean income	−2.403	−0.684	−0.412	−0.222	0.476	−0.479
Marital rate	−1.448	−0.651	−0.078	0.265	0.354	−0.221
Employment rate	−1.657	−1.07	−0.615	−0.144	1.122	−0.507

**Table 5 ijerph-19-10817-t005:** Pearson correlation significance test.

Air Pollutants	Pearson r	*p*-Value
Butadiene	0.119	0.083
Formaldehyde	0.093	0.175
Acetaldehyde	0.049	0.474
Benzene	0.245	0.000 *

* indicates p < 0.01.

## Data Availability

All data generated or analyzed during this study are included in this article. Further inquiries can be addressed to the corresponding author.

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
