# Peer review of "Spatiotemporal Correlation Analysis of Hydraulic Fracturing and Stroke in the United States"

_ijerph, 2022, doi:10.3390/ijerph191710817_

Round 1

Reviewer 1 Report

The authors aimed to evaluate the spatiotemporal correlation analysis of hydraulic fracturing and stroke in the United States. They found fracking ALE is moderately correlated with stroke mortality at ages over 65 in most states of fracking, in addition to cardiovascular disease and drug overdose being positively correlated with stroke mortality; the correlations between fracking ALE and stroke mortality in men appear to be higher than in women in several states. This is a well-written paper, which should be accepted in present form.

Author Response

The authors thank the reviewer for the positive comments. Thank you so much for your assistance.

Reviewer 2 Report

Dear authors

I have thoroughly read your manuscript Spatiotemporal Correlation Analysis of Hydraulic Fracturing and Stroke in the United States. That research has several merits, among which I include the fact that it was a pioneer in answering a very important public health question (Does fracking induce a higher risk of stroke?) and the geographic breadth of the study. The manuscript is well written, and follows the most important rule of scientific writing: “what we write about is complex, but the writing should be simple.”

Three other key points for good scientific communication were adopted throughout the manuscript: well supported, robust and reliable message and conclusions; appropriate and robust methodology and data analysis and clear explanation of how the article addresses existing knowledge gaps.

I am inclined to recommend it for publication to the editor of IJERPH, provided the authors make a few modifications to the manuscript, which I list below.

The message could be relevant to a wider audience. In the introduction, I recommend that authors include a paragraph that scales fracking oil extraction globally. This technique is used in how many countries? What is the world percentage of oil extracted by fracking? In my country (Brazil) a bill is being discussed to authorize fracking throughout the national territory (more information about it here: https://apublica.org/2022/06/governo-lancara-edital-para-estimular -fracking-no-brasil/), and evidently articles like yours will be important to support a debate on the pros and cons of this technology, here and in other countries.

In lines 58-60 you state that “To our knowledge, this article is the first attempt to address this question from a geographic information system (GIS) perspective using the extension of the geographically weighted regression (GWR) method”. Have you carried out a systematic review that corroborates this statement? In other words, what data do you rely on to include this statement in the manuscript?

In lines 232-233 the sentence “It demonstrates that fracking activity is not a significant factor that affects stroke mortality” sounded confusing to me. Is fracking activity not correlated with the incidence of strokes, by gender? This seems to contradict the statement in lines 237-239: “In contrast, the ALE coefficients for fracking for female stroke mortality in Montana, Wyoming, New Mexico, and Oklahoma are higher than those of other fracking states.”

In lines 234-239 it is unclear to me why: 1) The ALE coefficients of fracking for male stroke mortality around Marcellus shale (including Pennsylvania, West Virginia, and Ohio) are higher than those of other states of fracking; 2) In contrast, the ALE coefficients for fracking for female stroke mortality in Montana, Wyoming, New Mexico, and Oklahoma are higher than those of other fracking states.

In lines 249-253 you state that “It can be observed from the results of the GTWR analysis that fracking has a non-negligible effect on stroke mortality over 65 in most of the areas with fracking prevalent, as shown in Table 3 and Figure 3 However, its impact is relatively minor compared to other more dominant factors, such as cardiovascular disease and overdose.” Perhaps it would be advisable at this point in the manuscript to add a few sentences about the aging of the human population worldwide. My point is: in countries where fracking is allowed, wouldn't population aging make it unsafe for public health?

Finally, I would like to ask the authors which institution financed that research (lines 324-326): is the NSF a public or private institution? Does it have any relationship with or receive funds from oil companies? In that case, there would be a conflict of interest.

Reviewer 3 Report

the authors show a method to measure the population risk due to fracking wells in the people that live near fracking wells. They used  a double approach that considered time and spatial caractheristics. Stroke risk was investigated. In the tab 3 they showed that tobacco and High Cholesterol are protective regarding to ictus, but they did not  explain this result, in tab 4 only high cholesterol appeared  protective:  they should  discuss it. In the discussion (tab. 5)  the authors showed HAP monitoring station data, then they indicated the results as pearson correlation, but no concentration data was reported; they should add them. Moreover why this result would link to stroke risk, they do not discuss it.    

Round 2

Reviewer 3 Report

Ok, the new version is better than old version.

However you should discuss the tobacco and cholesterol data in the discussion, you writed it only in the figure; also you shoud explain why  "Benzene exposure has been shown to be associated with an increased risk of cardiovascular disease in previous study [69]" and then you retain it capable to increment stroke risk, instead you do not explain why it does not occur in the case cholesterol and tobacco.

A possible explanation could be that the age of cohort is high and this risks are less important, please add some reference that could support this idea.

Best regards
